# Public Libraries as Supportive Environments for Children’s Development of Critical Health Literacy

**DOI:** 10.3390/ijerph191911896

**Published:** 2022-09-20

**Authors:** Catherine L. Jenkins, Susie Sykes, Jane Wills

**Affiliations:** Institute of Health and Social Care, London South Bank University, London SE1 0AA, UK

**Keywords:** health literacy, children’s health literacy, critical health literacy, public libraries, settings-based approach, supersetting approach, supportive environments for health, social practice

## Abstract

Critical health literacy enables individuals to use cognitive and social resources for informed action on the wider determinants of health. Promoting critical health literacy early in the life-course may contribute to improved health outcomes in the long term, but children’s opportunities to develop critical health literacy are limited and tend to be school-based. This study applies a settings-based approach to analyse the potential of public libraries in England to be supportive environments for children’s development of critical health literacy. The study adopted institutional ethnography as a framework to explore the public library as an everyday setting for children. A children’s advisory group informed the study design. Thirteen children and 19 public library staff and community stakeholders were interviewed. The study results indicated that the public library was not seen by children, staff, or community stakeholders as a setting for health. Its policies and structure purport to develop health literacy, but the political nature of critical health literacy was seen as outside its remit. A supersetting approach in which children’s everyday settings work together is proposed and a conceptual model of the public library role is presented.

## 1. Introduction

### 1.1. Children’s Critical Health Literacy

Children’s access to and use of health information is influenced by their health literacy and the social contexts where they spend time, both with and without adults. This is because health literacy is a social practice [1,2] that has been shown to be a modifiable determinant of health [3] and an asset that can enable individuals to apply cognitive and social resources to support their own health and the health of their community from early in the life-course. Critical health literacy (critical HL) extends to planning, implementing, and evaluating interpersonal actions regarding the social determinants of health [4].

The life-course approach in health literacy research [5] has informed understanding of critical HL as developing alongside children’s cognitive and social maturation [6]. Functional literacy skills are not, however, prerequisite to the development of critical HL [7]. By age 10, children can be active critical HL practitioners [8,9], and there is a case for investing in developing critical HL early in the life-course to mitigate current and future burdens on health services [10,11]. Providing opportunities for children to develop their critical HL in primary school may pre-empt health-related misconceptions becoming resistant to change [12]. Children are already key public health actors [13] and health information brokers for older family members [14,15,16]. Some are also young carers with caring responsibilities [17]. During the initial months of the COVID-19 pandemic, children depicted their actions as protecting themselves, their families, and wider society [18].

Critical HL in children younger than secondary school age is under-studied. There is no definition specific to children’s needs [19], and the distinct circumstances whereby children’s opportunities to develop their health literacy ‘can be promoted or hindered by social structures, relationships, and societal demands’ [20] (p. 2).

The individual’s social context has been found to be important for the development of all health literacy [21] in the ways in which information is acquired and shared. Creating supportive environments is one of the action areas of the Ottawa Charter for Health Promotion, because of evidence that the everyday settings where children spend time, learn, and play, can influence their health [22]. The identification of supportive environments where critical HL can be developed by children is therefore a priority for research.

A scoping review by the first author to identify existing literature on critical HL in children returned 18 studies, 16 of which were school-based (the remaining two were co-located with schools). The review concluded that a supportive environment for children’s development of critical HL would have the following antecedents:acknowledges the wider determinants of health that matter to children;is open access and free at the point of use for children;involves children in how it is run;facilitates children’s informed actions for health.

These four antecedents, drawn from literature on the development of critical HL, provide the focus for analysis for this study of the potential of public libraries as a setting for its development.

### 1.2. Looking beyond School-Based Critical Health Literacy

School is where most children spend time. School-based settings—classrooms, whole-school assemblies, playgrounds, canteens—are frequently used for health literacy interventions targeting this population [23], as documented in frameworks like HeLit-Schools [24]. While there is some evidence that integrating critical HL into school-based health education may contribute to improved health outcomes in the long term [25], there are also well-recognised barriers to schools being supportive environments for children’s development of action-oriented critical HL. These barriers include schools’ structural hierarchies and lack of time and space to fully embed critical HL across the curriculum [26,27,28]. The purpose of the present study, therefore, is to explore potential non-school-based settings for promoting children’s critical HL.

Public libraries are one possibility. Public libraries are everyday settings with a core business tailored to the needs of the local communities they serve [29] and where children can access curated health information and signposting at no cost. Public libraries reach children like schools (and can also reach school-excluded children). In England, councils have a statutory duty under the Public Libraries and Museums Act 1964 to ‘provide a comprehensive and efficient library service for all persons’ who live, work, or study in the area [30] (there are no statutory requirements for schools to have their own libraries). Public library stock for children is recommended to be of sufficient range and quality to meet the social, developmental, educational, and leisure needs of all children and young people from birth to age 16, should promote information literacy, and should provide accurate and up-to-date information. Public libraries have a mandate for health promotion under the Universal Health Offer [31], which sets priorities for public libraries using a proportionate universalism approach: resourcing and delivering services to improve the lives of all, with proportionately greater resources targeted at the more disadvantaged in society to reduce the risk of inadvertently increasing health inequities [32]. The wider Universal Health Offer framework includes a pledge to support children’s health and well-being and the Children’s Promise [33], and case studies of universal offers in practice include public libraries working in partnership with the National Health Service (NHS), local public health departments, and universities [34]. These joint efforts constitute more than ‘health promotion in a setting’ [35]: they are part of a concerted settings-based approach. Public libraries require further study as potential supportive environments for children’s development of critical HL.

## 2. Materials and Methods

### 2.1. The Setting

The study took place in a public library system in the East of England comprising over 40 individual public-access static and mobile branches. Some branches are co-located with other types of settings, e.g., a school, health clinic, or sports centre. Prison-based branches were excluded (because these are not accessible to children), and the system’s Schools Library Service had ceased operations prior to the study. The system operates on an Industrial and Provident Society (IPS) model, which is required under legislation to be run for the benefit of the community [36]. COVID-19 risk assessments and additional training on remote research methods were completed prior to accessing the setting in its physical and online forms. The study received ethics approval from London South Bank University (ETH2021-0003).

In order to understand how the library is used, in-depth insights were needed into the experiences of the children, staff, and stakeholders in situ. The study therefore drew on methods from institutional ethnography (IE) to guide iterative data collection and analysis, including ‘looking at documents, talking to people and watching their work’ [37] (p. 349), to understand how people’s work is socially organised (where ‘work’ is understood as any activity done with purpose that takes time and effort) [38]. IE is well suited to settings-based research, and its toolkit proved adaptable to the remote research conditions necessitated by the COVID-19 pandemic.

### 2.2. The Informants

The study deployed purposive and snowball sampling to recruit 13 child informants (with the consent of their parents/caregivers), 13 public library staff informants, and six community-stakeholder informants. It is best practice in IE to refer to participants as ‘informants’ because their knowledge and experiences actively inform the IE researcher’s understanding of the setting [39]. Child informants were recruited through a poster campaign designed in consultation with a panel of child advisors who were not involved in the study as informants. The poster campaign was shared on social media (Facebook and Twitter) visible to parents/caregivers. Copies of the poster were also attached to an email sent to the local public health department for circulation. Recruitment priorities were guided by information power [40], meaning that informants’ diverse and first-hand experiences relevant to the topic and processes being studied were prioritised over a theoretical saturation threshold [41]. Child informants were aged between seven and 11 years and in primary school education. These criteria were set in response to the demonstrated lack of health literacy research involving ‘children under the age of ten or within a primary school context’ [42] (p. 21). This age range falls within middle childhood, a foundational period of independent decision-making and the formation of health beliefs after which, it has been claimed, it may be ‘too late’ (at the onset of adolescence) to begin developing health literacy [43] (p. 632).

Of the 13 child informants who participated, seven were interviewed in their local library branch and were regular visitors there. The remaining six were interviewed online, of which three had access to a library membership card (their own, or that of their parent/caregiver). Seven child informants chose to be interviewed collectively in three sibling/friendship groups. Child informants selected their own pseudonyms, or “research code names”.

Public library staff informants represented a mix of job roles (e.g., property manager, library and information advisor, stock librarian) and locations in the organisational chart, generally split between frontline work in the Service Delivery team and back-office work in the Content and Resource Development team. Community-stakeholder informants were recruited to provide extra-local perspectives on the setting’s work and included a library design consultant, IPS trustee, business improvement manager (from the local council), and staff from a local NHS hospital-based library.

### 2.3. Children’s Involvement and Engagement

There are not many precedents in the literature of children’s participation in health literacy research. For this reason, it was important to consult with children to ensure that the research focus was relevant to them, particularly in the context of the other demands on children’s time during the early waves of COVID-19.

The study design, including its ethics, informed consent documentation, data collection, and plans for dissemination, were consulted on with a panel of eight child advisors (CAs). The CAs were in the same age range as the child informants and contributed to the research via videoconferencing and post. Of the CAs, two used a school library but not a public library, four were not members of the public library but used it occasionally via their parents’/caregivers’ membership card, and two were card-carrying members of a public library.

The involvement of the CAs and the child informants followed the principles of the Lundy model of participation [44] from initial rapport-building through to final evaluations of children’s experiences of participating in a remote research project [45]. The Lundy model guidelines helped create conditions under which it was unacceptable for the researcher to solicit children’s views and then fail to take those views into account. All CAs and child informants received a certificate of participation and a tote bag containing materials to conduct their own research projects (a notepad and pens), a leaflet signposting children to public library-based health resources [46], and a middle-grade fiction book with a storyline that showed children’s critical HL in action [47].

### 2.4. Data Collection and Analysis

In IE, texts are understood as governing or mediating people’s work in the setting. Texts are defined multimodally and can include policies, drop-down menu options, photographs, social media posts, and any other media replicable across and beyond the setting. Texts are actioned by people, including texts unseen in their originals. Public library staff and community stakeholders were invited to share their knowledge and work related to children’s critical HL in semi-structured text elicitation interviews. IE adopts a flexible approach to its topic guides for interviews and observations. Questions were led by the texts that staff and community-stakeholder informants chose to bring along to their interview or to which they referred during their interview.

The interviews with the child informants used child-generated drawings and a modified ‘interview to the double’ (ITTD) technique [48,49] to elicit in-depth explanations of what work children do in the public library when they want to find out about health or engage in health advocacy. The aim of ITTD is that the informant conveys ‘a day in their life’ account of their experiences in the setting to the interviewer in detail of sufficient specificity that the interviewer could plausibly replace them at work the next day as their body double or doppelgänger.

The ITTD technique formed part of a critical HL activity that asked children to draw and describe as if they were presenting a livestream on YouTube Kids or TikTok for Younger Users how a public library setting on Earth could support an alien to take informed action for alien and human health. The activity encouraged children to follow their curiosity and wander around the public library branch or, if online, the library web pages. The substitution of an alien for the double, to create an ‘interview to the alien’ (ITTA), was suggested by the CAs as a playful way to redress the power imbalance between the adult researcher (for which the alien is a proxy) and child informants. Child informants were invited to create their own alien. Aliens have previously been used in critical HL research with children [8], and can bring value by situating children as knowledgeable, in contrast to the alien, who knows very little and is reliant on children sharing their experiences.

Throughout data collection and analysis, leads for inquiry were pursued on an ongoing basis in response to the researcher’s growing knowledge of the social context of the setting and people’s activities there. Multimodal data (texts collected during site visits, e.g., researcher-generated photographs of signage; texts elicited from staff and community-stakeholder informants; and interview transcripts captured by Otter.ai and manually checked by the researcher) were uploaded to NVivo 12. Using NVivo helped structure the analysis, firstly by de-familiarising the exported data and secondly by facilitating keyword- and tag-based querying across the dataset as a whole. Neither coding nor thematic analysis was used, because IE avoids reproducing abstract concepts that might obscure the work that people do [50]. Instead, the four antecedents to a supportive public library setting for children’s critical HL, as identified from the literature, provided the theoretical framing and guided the analysis by sensitising the researcher to possible lines of inquiry to pursue.

In IE, indexing is used to organise data. The antecedents provided the first entries in this index and were expanded and cross-referenced with further entries and sub-entries from the data. Once organised into an index, the data were analysed using an abductive approach to pinpoint relevant empirical evidence that was ‘surprising’ or ‘puzzling’ when viewed through the analytic lens of the antecedents and in the context of the setting. Pursuing the lines of inquiry that opened up from these led to the insights that structure the Section 3.

## 3. Results

The results are reported under the insights about this setting that they provide evidence for. These insights are selected from the macro-, meso-, and micro-levels of the setting.

### 3.1. The Public Library Is Not Perceived as a Setting for Health: “It’s More Signposting […] without Going That Step Further”

Child informants tended to view health as incidental to the public library setting, not core to it. From their perspective, the library as a setting for health was limited to provision of contemporary public health measures, such as the COVID-19 test kits available at the public library entrance:

Well, they’ve got [COVID-19] testing where you just do the nose. They’ve got that. (Child informant, code name: Ice Cream)

Despite public libraries’ role in the provision of consumer health information being a live issue in the UK and farther afield [51,52], staff informants similarly did not connect their work, or their workplace, with children’s health:

It’s not common for a child to ask about health. (Library and information advisor, Service Delivery)

I’m sorry [the interview] wasn’t necessarily super health-based. I don’t, I don’t know that we offer that many specifically health-based things here. (Assistant library manager, Service Delivery)

Children were viewed by staff as reluctant to draw attention to their health-related concerns in this setting, even while the library was upheld by staff as a ‘safe place’ for children wary of scrutiny:

They [children] don’t want to do anything [that] may potentially cause problems in terms of social services or there’s all these sorts of worries that a lot of young carers and things have as well […] Takes them maybe a long time, but let[s] them realise that this is a safe place that they can come to. (Library manager, Service Delivery)

Health-related publications specifically for children that were available in the library at the time of the study included one [53] that featured on the Reading Agency’s COVID-19 edition of its Reading Well for Children booklist [54]. The book that was most frequently referred to and sought out by children in this study, however, was not on either of the Reading Well booklists [55].

There are factors contributing to the public library setting not being seen as a setting for health. One is the ‘health by stealth’ approach to health promotion in the public library setting, which “tends to happen around special days” (library and information advisor, Service Delivery) booked on a sector calendar. On other days, it is subtle by design:

We don’t actively badge it as Children’s Promise or Reading Well [health-related signposting] for children […] a customer wouldn’t know, necessarily, that they were being steered towards particular books […] It’s all very ‘stealth’. (Executive library manager, Service Delivery)

Another factor is the inconsistent representation of public libraries’ health-related remit in texts referred to by staff for benchmarking service provision: the Universal Health Offer [31] and the Children’s Promise [33].

Different versions of the Universal Health Offer co-exist on the website of Libraries Connected, the advocacy organisation for the sector. The multiple versions reflect relaunches of the framework over time, most recently for the recovery of public library services in the context of the ‘new normal’ of living with COVID-19. Against this backdrop, the Universal Health Offer lands in the public library setting as vast in scope:

It is quite hard to, like, where do you choose, like, *which* health, you know, it’s not like one part, I often think, Oh there’s so many different conditions that, y’know, we should give more attention to. (Information for Living librarian, Content and Resource Development)

If we pull together all our knowledge and resources, everything we could offer, staff could then promote to children […] it’d be good to maybe actually have a, have a little umbrella module developed, which is what can we do to offer support to children. (Information for Living librarian, Content and Resource Development)

The Children’s Promise provides guidance for staff on operationalising the Universal Health Offer to ensure children ‘benefit from targeted library service activities that address disadvantage and improve their health and wellbeing’ [33] (p. 1). The text of the Promise borrows from life-course discourse in public health by mapping out a ‘library journey’ that parallels children’s developmental stages, but makes no provision for making children aware that they are on this journey: the text is not visible from where children stand in the public library context. The current Children’s Promise makes no promises of health literacy support for children.

Staff informants described work to develop children’s awareness of the determinants of health as hindered by the lack of published resources available to support such work:

There just isn’t the, the stuff there […] I do wish there was, um, a child-friendly place we could direct them to […] part of our role is making sure people know where to find the right information. And when the information isn’t there to *be* found, at the level it needs to be at, it’s difficult. (Stock librarian, Content and Resource Development)

The concept of health literacy was taken by staff informants to mean functional health literacy and used synonymously with signposting: guiding people towards information, but ‘without going that step further’ towards taking action on the determinants of health that critical HL entails:

I think to a degree, it [health literacy] is sort of in the job description. But I think it’s more as I say it’s signposting. And it’s ensuring that you know where the information is to support that child, that parent. Um, without going that step further. (Library manager, Service Delivery)

### 3.2. Schools Are Key Partners for Children’s Access to the Public Library System: “Get Them in the Door and That’s Usually through Schools”

Staff informants frequently referred to relationships with local schools as providing routes through which children were introduced to the public library setting:

Before COVID, we would have regular class visits in, so we worked very closely with one of the primary schools. (Library manager, Service Delivery)

Schools are referrers of children into public library settings, and ‘schools, school library services and school librarians’ are named first in the list of partners for children’s ‘library journey’ in the Children’s Promise [32] (p. 1). Schools funnel children through the physical and digital public library doors:

Just initially get them in the door. And that’s usually through schools. (Library manager, Service Delivery)

Children ‘becoming part of the library culture’ (stock librarian, Content and Resource Development) is enshrined in the Children’s Promise: ‘[children] should be actively involved in decisions about library service developments’ [33] (p. 1). The redesign at one branch of the children’s area of the library in partnership with a library-design firm contrasts children’s opportunities to participate in this setting with school-based opportunities:

We operate in public libraries, and also in schools […] schools might have actually sat down and consulted with their School Council, or, y’know, Year Six, or whatever it happens to be, and had some input from the children themselves. (Stakeholder informant, library-design consultant)

Children’s involvement in health-related work tended to be outsourced by the public library system:

Lots of the work that we do is partnership-based […] if it’s not appropriate for a member of library staff to kind of do something around mental health and well-being, can we get an expert partner in […] we can provide some kind of access to expertise in the community […] so we could look at bringing in, um, yeah, bringing in the expertise […] we could bring in other charities, other partners. (Well-being manager, Service Delivery)

Health literacy work targeting children entailed bringing in external expertise (as in the staff training provided by the NHS library team).

### 3.3. The Public Library System Seeks to Differentiate Its Offer from That of Schools: “We Don’t Work Like That”

At the same time as working with schools, the public library system seeks to distance itself from school-based ways of working. Learning in the public library aims to be ‘distinct from the school offer’ [56] (p. 10). The self-assessment checklist for auditing the Children’s Promise lists ‘learning spaces in the children’s area where they can learn individually and or in formal/informal groups’ [57] (p. 14). Children’s library-based learning opportunities, as distinct from school-based ones, were discussed by staff as libraries’ unique selling point. As one staff informant emphasised, libraries “don’t work like” schools:

So we’re not actively, y’know how school is—You must read this, and you need to do this […] We don’t work like that. (Executive library manager, Service Delivery)

Last year [2020], there was, where it went all online, there was a massive dip in take-up, because we found that children like coming in, they like coming in and talking to a member of staff […] they like having that engagement, and doing it online just took all of the, the joy out of it. And I wonder if it also made it a bit like schoolwork. You’ve got to read this book and then you’ve got to go online and you’ve got to fill out the thing. Whereas if you come in and talk to somebody, you’ve got that interaction, you’re going to choose some other books, you might bump into your friends, perhaps it’ll turn into a spontaneous playdate […] it’s that added value. (Executive library manager, Service Delivery)

The relationship between the public library system and schools was less about one offering an “alternative” to the other than it was about how both are interdependent.

### 3.4. Age Limits Children’s Access to and Use of the Public Library System: “They Don’t Want Random Children Just Running into the Library”

School-based settings matter for whether public libraries can be supportive environments for children’s critical HL, because both settings are required to coordinate with each other as part of a supersetting approach. As a factor that limits children’s access (one of the antecedents) to settings outside schools, age can also be traced back to schools’ influence, because public libraries’ age-based access policies take their lead from schools.

Staff informants stated that work was organised on the assumption that older children’s needs were already covered by schools: “Yes, there’s less [public library services] for seven to 11-year-olds. But that’s because they’re in school” (library and information advisor, Service Delivery). Specific support for children’s critical appraisal of health-related information, a dimension of critical HL, was limited to a setting-wide subscription to an online encyclopaedia with age-based login access that provided filtered information on health topics.

Tracing funding pipelines showed that making health-related services available for seven to 11-year-olds requires workarounds to redirect funding actually ring-fenced for adults, families, or younger siblings:

Because I’m adult mental health-funded, there’s only so much like young people, children stuff that I can sort of get away with […] there are some ways we can get around it. So we’ve had some funding around families and carers. Um, we have our perinatal service […] And many of those parents also have more than one child. So there are ways that that kind of supporting children and young families kind of trickles through what kind of core funding allows us to do. (Well-being manager, Service Delivery)

Age matters for children’s access to and use of this setting. The Children’s Promise text maps children’s life-course along a timeline of transitions from one school-based setting to another. Access to the setting and its resources is experienced by children differently, depending on where along the timeline children are located by staff, whether staff define children as customers, and whether staff engage in safeguarding work.

Children showed awareness in interviews that their access to the public library setting improved if they were accompanied by an adult: “Because they don’t want random children just running into the library’ (child informant, code name Ginny Weasley). Children’s access to the public library was subject to gatekeeping and safeguarding by adults. Monitoring children’s access was built in to the physical setting design:

We’ve changed this [children’s area] all round physically so that we can see what’s going on. It was a very different space when I came. There were a lot of blind spots. And that’s something, that is for, for my safety but also for the users’ safety as well […] There’s still a couple of blind spots but our head of finance has given me the OK to buy some of these corner mirrors […] if we’re comfortable, then we are going to be relaxed and welcoming to chil—the boys and girls that, y’know, may potentially need some support. (Library manager, Service Delivery)

Staff were, however, trusted by senior management to use their professional judgement in regard to age-restricted access:

And they can come in on their own and it’s okay—they’re not going to be questioned […] Where’s your adult, y’know […] if you’ve got a very young one, then obviously, but by the time they’re eight, nine, ten, it’s okay I think for them to be coming in and left on their own. (Library manager, Service Delivery)

Staff had licence to use their discretion on whether to put age-based policies into practice.

### 3.5. Legislation Regulates the Appropriateness of Public Library Services: “We Can’t Be Seen to Be Involved in Anything That Might Become Political”

The extent to which the public library setting could support children to address the wider determinants of health was constrained by political considerations and adults’ ideas of what constituted appropriate library-based activities.

Child informants’ suggestions on how the setting could support them to take action on health were framed in terms of what would be ‘allowed’ in the setting:

They could help you maybe like, help you get it [a health-related call to action], get it ready, so that you can like show it, or something. Or help you make the poster if you were doing a campaign or something […] maybe stuff up on the wall or stuff on the tables or in books that tell you what is happening right now. At this moment. And what. If you’re allowed in the library. What you can do in the library and stuff. (Child informant, code name Ginny Weasley)

Child informants’ other suggestions included having “a little area” (child informant, code name Nicolai) or “critical corner”, where resources and inspiration for action could be accessed in one place, rather than being “a bit jumbled among the non-fiction” (child informant, code name Nicolai).

By-laws regulating the use of the public library under the 1964 Act [30] set out appropriate standards of behaviour for library users and the actions to be taken if such standards are contravened. By-laws limited what staff informants reported they were able to do in this setting to support children’s awareness of and action on the determinants of health:

In terms of activist and activism and being involved in that, we have to be quite careful as an organisation, um, we can’t be seen to be involved in anything that might become political. So our by-laws and things restrict us from having petitions and […] campaigns and those kinds of things in our spaces […] we have to balance, we have to be there for everybody. And we have to be politically neutral, and we have to be unbiased […] yeah, it’s a bit tricky that one […] particularly as, y’know, we move into election periods, and um we have […] we have, y’know, to be quite careful in what we do and don’t have in in the library space. (Executive library manager, Service Delivery)

By-laws manifested operationally as concerns about the appropriateness of the public library as a setting for children’s critical HL development:

We’re in such a unique place, I think it’s important to remember that first and foremost, we are a library service. And there’s only so much that we can do that’s appropriate […] it’s quite a delicate balance between what we can do and what’s appropriate for us to do […] But what we can do is make sure that the community has access to the best, most up-to-date resources and books and people to talk to. (Well-being manager, Service Delivery)

However, strict adherence to by-laws was, like the enforcement of age-based restrictions, at individual staff discretion:

It’s down to common sense what we would enact and use […] Mostly, we’ll only kind of apply a few of [the by-laws], as and when they’re needed. (Executive library manager, Service Delivery)

We would find MPs’ addresses, we would do all that kind of stuff in the same way as we would enable anyone wanting to do anything that needed assistance doing it. (Stock librarian, Content and Resource Development)

Operating unbounded by the by-laws was possible at the micro-level of individual branches.

## 4. Discussion

The results of the study are twofold. Firstly, it is only at the micro-level of the public library system (individual library branches) that all four antecedents to a supportive environment for critical HL (see Figure 1) were evidenced in practice. Secondly, a public library-based approach to children’s development of critical HL is not enough on its own: a settings-based approach at the public library must form part of a wider supersetting approach, i.e., complemented by (not alternative to) other settings, including schools.

The supersetting approach, or “settings-based approach 2.0” [58], is a multi-settings approach to health that recommends ‘coordinated activities […] carried out in a range of different settings within a local community with the aim of attaining synergistic and sustainable effects’ [35] (pp. 30–31). The model shown in Figure 1 conceptualises the public library system as part of a supersetting approach that involves the participation of children and other settings in partnership towards a common goal of multilevel action on the wider determinants of health. The supersetting approach recognises that health is created across settings acting in concert with one another. For children’s development of critical HL, a supersetting approach could look like a public library system working alongside schools and other settings, such as youth clubs.

At the macro-level, the social context in which the public library system operates influences the extent to which it can offer a supportive environment for children’s development of critical HL. This wider context includes governance of the sector (e.g., by-laws) and other settings, especially schools. Within the library, meso-level factors, such as individual library staff expertise and attitudes, individual branch infrastructure and layout, target group-specific funding, and stock and resources, all influence the support available for children’s development of critical HL.

The micro-level is more conducive (for all four antecedents) to a supportive environment for children’s critical HL. It is at the micro-level that a library may organise groups that enable children to become locally involved. This is shown in Figure 1, which represents the public library as part of a supersetting approach.

The development of critical HL in children is an under-researched area, and it is known to be difficult to develop [43]. The few studies available tend to be school-based and limit critical HL to critical appraisal skills, with less emphasis on informed action for societal and population health [59]. The COVID-19 pandemic has highlighted the importance of children’s critical HL in their roles as public health actors outside school-based contexts [18]. The importance of this study, therefore, is that it identifies settings other than schools for children’s development of critical HL.

The conceptual model contributes to the literature on the settings-based approach by highlighting the need to pay more attention to the wider (macro-level) context in which settings operate and the potential of group activities, organised at the micro-level of individual library branches, to offer local workarounds. The settings-based approach to health promotion aims to embed health into the core business of a setting, such that it becomes organisationally normalised and ‘the way things are done around here’. Previous research has pointed out that a successful settings-based approach is difficult to evidence, because health embedded into core business recedes into infrastructure and becomes taken for granted [60]. What the conceptual model contributes is a cross-section of a supersetting approach involving the public library system that can be used to trace how macro-level political and extra-local dynamics interact with bottom-up interests to move the public library ‘beyond a relatively limited “information provision” model’ [29] (p. 899) and towards a supportive environment for action-oriented critical HL. The public library system is positioned along a lifespan/setting continuum [61] that—in the model—becomes a life-course approach combined with a supersetting approach. A combination of approaches is required for the potential of the public library system as a supportive environment for early life-course critical HL development to be reached.

The study has implications for how the public library system as a setting for health is understood, and consequently priorities for future research and practice. Whilst settings are accepted as part of the global approach to promoting health, there has been little development beyond education and health sectors. A recent handbook [62] includes examples of non-traditional and emerging settings for health, such as airports [63]. Digital environments and social media are increasingly recognised as settings for health [64,65]. As hybrid (physical and digital) settings, public libraries are well placed to contribute to research and practice in this area.

Follow-up studies indicated by the results include further consideration of the settings-based approach in health literacy research and greater involvement of children in critical HL studies in ways that take into account children’s social contexts and learning from multidisciplinary insights into participatory research, e.g., from library and information science [66]. The literature on the settings-based approach to health promotion and health literacy must continue to adapt to and absorb twenty-first-century settings into its research priorities [62], and further research building on public libraries’ potential as everyday settings for health [29] and health literacy [67] should be conducted.

### Limitations

This study was conducted during COVID-19 as part of a doctoral research project and was constrained by children’s availability for interviews at a time when parents/caregivers were under additional pressure, either from homeschooling or managing quarantine periods following travel abroad.

## 5. Conclusions

Public library systems are a statutory requirement in England, and are obligated to provide services supportive of the health of local communities. They are therefore key everyday settings to which most children have access, and their inclusion in a supersetting approach with schools could offer one solution to the problem of embedding critical HL in school curricula. Joining public library systems and schools together so that the two can work in synergy with each other could help overcome the structural barriers to action-oriented critical HL present in both when each is viewed in isolation.

Critical HL is a social practice developed in response to the resources at hand and embodied knowledge [2]. Children should be supported to draw upon and relate critical HL learning to their pre-existing, contextualised understandings. Neglecting to do this may risk exacerbating health inequities, because children who struggle to reconcile public health messaging with their everyday social contexts may also struggle to put such messaging into practice [8]. Overemphasising one setting to which children have access (schools) over other everyday settings curtails possibilities for how those settings might work in concert with each other to ensure a joined-up approach to developing critical HL earlier in the life-course.

This study has inquired into what makes a setting—as the Ottawa Charter [22] understands the concept—a supportive environment for critical HL development. The study concludes that the same ‘cautious optimism’ applied to the connection of school-based health education to critical HL in the long term [25] (p. 13) can also be applied to a supersetting approach that combines schools with public library systems as supportive environments where children can create, critique, and take control of their health and that of others.

## Figures and Tables

**Figure 1 ijerph-19-11896-f001:**
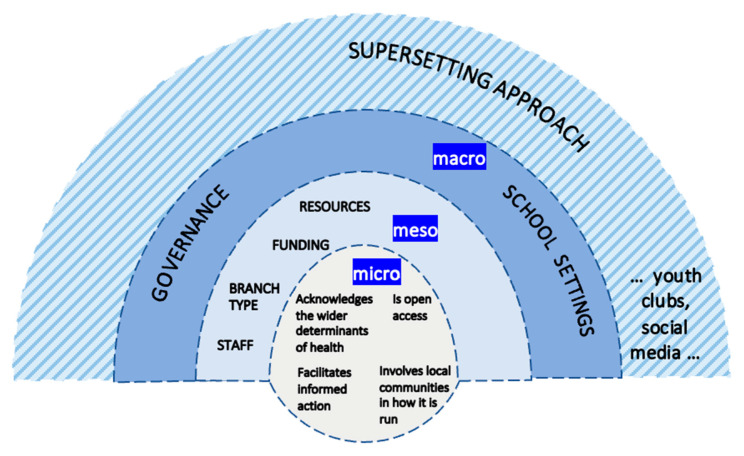
Conceptual model of the public library system as a supportive environment for children’s development of critical health literacy.

## Data Availability

Not applicable.

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
