# Peer review of "Public Libraries as Supportive Environments for Children’s Development of Critical Health Literacy"

_ijerph, 2022, doi:10.3390/ijerph191911896_

Round 1

Reviewer 1 Report

Please see comments on the attached word doc.

Reviewer 2 Report

The manuscript presents an interesting and relevant topic for the health, educational and social fields.

However, it would be recommendable to strengthen the theoretical framework with the administrative policy that regulates the provision of bibliographic resources to libraries. It would also be useful to know what health-related publications are available in the libraries under study. This data would make it possible to evaluate the potential of these institutions for health education.

It would also be interesting to know the instrument used in its entirety, in order to see in greater detail what type of questions have been asked and to better evaluate the results.

Round 2

Reviewer 2 Report

Although suggestions for improvement have been only partially addressed, the manuscript is sufficiently grounded at the theoretical and methodological aspects.